# An Efficient and Universal Protoplast Isolation Protocol Suitable for Transient Gene Expression Analysis and Single-Cell RNA Sequencing

**DOI:** 10.3390/ijms23073419

**Published:** 2022-03-22

**Authors:** Juanjuan Wang, Yang Wang, Tianfeng Lü, Xia Yang, Jing Liu, Yang Dong, Yinzheng Wang

**Affiliations:** 1State Key Laboratory of Systematic and Evolutionary Botany, Institute of Botany, Chinese Academy of Sciences, Beijing 100093, China; jwang2@mpipz.mpg.de (J.W.); wangyang2017@ibcas.ac.cn (Y.W.); lvtianfeng@ibcas.ac.cn (T.L.); yangxia@ibcas.ac.cn (X.Y.); liujing2010@ibcas.ac.cn (J.L.); 2University of Chinese Academy of Sciences, Beijing 100049, China

**Keywords:** protoplast isolation, cell walls, *Chirita pumila*, protocol, transient gene expression, single-cell RNA sequencing

## Abstract

The recent advent of single-cell RNA sequencing (scRNA-seq) has enabled access to the developmental landscape of a complex organ by monitoring the differentiation trajectory of every specialized cell type at the single-cell level. A main challenge in this endeavor is dissociating plant cells from the rigid cell walls and some species are recalcitrant to such cellular isolation. Here, we describe the establishment of a simple and efficient protocol for protoplast preparation in *Chirita pumila*, which includes two consecutive digestion processes with different enzymatic buffers. Using this protocol, we generated viable cell suspensions suitable for an array of expression analyses, including scRNA-seq. The universal application of this protocol was further tested by successfully isolating high-quality protoplasts from multiple organs (petals, fruits, tuberous roots, and gynophores) from representative species on the key branches of the angiosperm lineage. This work provides a robust method in plant science, overcoming barriers to isolating protoplasts in diverse plant species and opens a new avenue to study cell type specification, tissue function, and organ diversification in plants.

## 1. Introduction

A central question in developmental biology is understanding how the information on DNA sequences are translated into physiological and morphological traits along with the differentiation of stem cells [1,2]. During post-embryonic development, organs composed of distinct tissue with unique functions are continuously produced from the stem-cell derived tissues, called the shoot apical meristems (SAM). In the process of organ development, tissue patterning is modulated by the dynamic and spatio-temporal changes in the expression of “toolkit” genes that specify cell fate and regulate cell differentiation [3,4,5]. Variations in expression patterns of these conserved “toolkit” genes underlie most of the reported diversity of organ shape and function [6,7,8,9,10,11]. Therefore, understanding the molecular mechanisms governing the stepwise cell fate specification holds the key to deciphering the principles that regulate organ patterning during plant development and to unlock the genetic basis of natural diversity and adaptation.

The recent advances in single-cell RNA sequencing (scRNA-seq) have revolutionized the studies in plant developmental biology and provided an unprecedented opportunity to access the differentiation trajectory of plant organs at single-cell resolution [12,13]. For a successful scRNA-seq experiment, the prerequisite is to generate high-quality and viable cell suspensions, which relies on breaking down the cell wall matrixes using hydrolytic buffers from tissues of interest [14]. However, the plant cell wall differs in thickness and composition depending on plant species, tissues, developmental stages, and environmental conditions [15]. This unique feature of plant cell creates a hurdle for protoplast preparation, which is therefore limited to model systems, such as *Arabidopsis* and rice, which have a stable and well-developed protoplasting protocol [16,17,18]. In the past decades, various protoplast isolation methods were described in non-model systems, such as pineapple, switchgrass, cucumber, wheat, and orchids for transient expression and biochemical analysis [19,20,21,22,23,24,25,26,27]. While these studies present significant progress in development of methods in protoplast isolation from diverse plant species, they also highlight the caveat in comprehending the scRNA-seq data using species-specific enzymatic buffers to generate single cell suspensions as the protoplasting process triggers substantial transcriptional responses. In addition, the efficiency and usage of the above-mentioned methods in different taxa remain to be determined. Taken together, all these factors spark the idea to develop a universal protoplast-preparation method, which enables the wide application of this technology among species to the elucidation of the developmental pathways responsible for morphological diversification and environmental adaptation.

Here, we report an efficient two-step digestion protocol for protoplast isolation in leaf mesophyll of *Chirita pumila*, which is a newly developed model system for studies on floral organ development. We showed that the protoplasts prepared from this method are viable and competent for transformation, and the transformation rate is further optimized to 60–70% upon heat-shock treatment. We further tested the utility of this protocol from diverse organs of multiple species representing key lineages in the angiosperms. Using this protocol, high-yield and -quality protoplasts were recovered from diverse organs of different species. Therefore, the protoplasts isolation method described here will be particularly useful in probing the cellular basis of organ development and in tracking the molecular pathways underlying the morphological novelties.

## 2. Results and Discussion

### 2.1. Establishment of an Efficient Protoplast Isolation Method in C. pumila

Unlike animal cells, plant cells are surrounded by a rigid and semi-permeable structure: the cell wall. The cell wall determines the final shape of the plant cell and is composed of a complex matrix of polysaccharides [28]. Pectin is the major adhesive material between cells in the middle lamellae between the primary cell walls, where it forms a gel-like network together with hemicellulose and other low molecular weight proteins [29,30,31]. The pectin matrix provides a skeleton for the deposition and extension of the organized crystalline microfibrils, which determine the cell wall characteristics [29,32,33]. Therefore, the breakdown of the pectin-associated matrix with pectinase is crucial for the efficient dissociation of plant cells from the tissues and is a prerequisite for efficient protoplast production [20,34,35].

In this study, we systematically examined the effect of pectinase in combination with different ratios of cellulase/macerozyme. In the case of *C. pumila* leaves (Figure 1a), we found that digestion with enzyme buffer containing 1% cellulase, 0.5% pectinase, and 0.5% macerozyme resulted in the highest protoplast yield within 3–4 h (Table 1). However, the protoplasts generated by this buffer showed incomplete digestion and the unbalanced osmotic pressure during enzymolysis. In maize, pretreatment of the samples with the balanced osmotic buffer prior to digestion significantly increases the efficiency of protoplasts generation [36,37]. We therefore applied a pretreatment buffer to the *C. pumila* samples under vacuum infiltration for 10 min. We found that the stability and activity of protoplasts was significantly increased from 78.01 ± 3.20% to 92.97 ± 1.43% with the pretreatment (Figure 1b,c and Appendix A). We next determined whether secondary digestion could increase the efficiency of protoplast isolation from incompletely hydrolyzed tissues. Secondary digestion with a buffer containing 1.2% cellulase and 0.4% macerozyme for an additional 60–90 min effectively dissociated the tissues and increased the yield of protoplasts.

The enzymatic digestion buffers contain chloride and sodium, which will inevitably stress the cell and generate global transcriptional inductions for genes involved in stress responses and turgor maintenance [38]. It should be noted that the effect of protoplasting on the transcriptome must be filtered out from the scRNA-seq data by a sister RNA-seq experiment, comparing the isolated protoplasts and the undigested tissues [13,17]. However, cautions should be taken for developmental genes, whose expression is dynamically regulated by conditional epigenetic modifications [39,40,41]. In order to evaluate if the protoplast production process results in a global epigenome alternation, we monitored the RNA level of *CpHDA6*, *CpHDA19*, *CpIBM1*, *CpSWN*, and *CpCLF*, which are the key executive factors involved in the histone modification machinery [39,40,41,42,43,44]. We found that none of these genes exhibit significant changes between the protoplast and the intact leaf samples in *C. pumila* (Appendix A). Therefore, unlike the substantial epigenetic reprogramming induced by wounding in the tissues [45,46], the protoplasting process does not generate significant transcriptomic fluctuations that result from epigenetic remodeling.

The main goal of an optimized protoplast preparation method is to produce a clean cell suspension with high cell viability. We next turned to testing the quality of the cells by monitoring the cell viability with time-lapse (1 h, 1.5 h, 2.5 h, 3 h and 12 h) fluorescein diacetate (FDA) staining. As shown in Figure 2, we found that ~89% of the cells in the protoplast suspension are viable, as indicated by their round shape with strong fluorescent signals (Appendix A). This result was further substantiated by the time-lapse Rhodamine 123 staining [47] with most of the cells exhibit fluorescent signals in the corresponding time point (Appendix A).

In conclusion, the method with two consecutive digestion processes developed in *C. pumila* generates viable cell suspensions without significant epigenetic remodeling, which is suitable for downstream experiments, such as scRNA-seq and transient expression analysis (see below).

### 2.2. Development and Optimization of the Protoplast Transformation Platform for C. pumila

In order to test if the protoplasts isolated from the above protocol are competent for transformation, we first resorted to the polyethylene glycol (PEG)-CaCl_2_ mediated plasmid transformation, as it is a widely-used and well-developed approach [19,48,49,50]. Unfortunately, by monitoring the green fluorescent protein (GFP) expression in the protoplasts, we found that the transformation efficiency was only ~7–10% when using the conventional 40% PEG4000-mediated transformation (Figure 3a), indicating further optimization was required to increase the efficiency.

Cell membrane composition and temperature are the two determining factors affecting cell membrane fluidity, and heat shock can increase the fluidity of the cell membrane to facilitate the absorption of exogenous DNA [51,52]. In the case of mesophyll protoplasts of *C. pumila*, we found that heat shocking at 42 °C for 3 min mildly increased the transformation efficiency from ~7% to 20% (Figure 3a,b). The transformation efficiency was increased further to ~60% when the concentration of CaCl_2_ in the PEG solution was altered to 0.2 M and WI solution was substituted by W5 in protoplast culture (Figure 3c,d). In orchids (*Cymbidium*), it was shown that the amount of plasmid used for transformation had a profound effect on the transformation rate [53]. We therefore assessed whether altering the plasmid quantity could further enhance the transformation efficacy. We gradually increased the plasmid quantity from 2.5 to 20 μg and the transformation efficiency gradually increased and peaked at ~70% with 20 μg plasmid (Figure 4a–d). Thus, the transformation efficiency of the *C. pumila* mesophyll protoplast under the optimized parameters is comparable to that in the *Arabidopsis* model system [19].

We then aimed to verify whether the protoplasts are active for transient expression analysis. In *Arabidopsis*, ABSCISIC ACID–RESPONSIVE ELEMENT BINDING PROTEIN1 (AREB1) binds to abscisic acid (ABA)–responsive element (ABRE) motif in the promoters of AP2-family members to activate the transcription of the target genes [54]. We therefore cloned the promoter of *C. pumila TOE3* (*CpTOE3*), homologous to the *AtAREB1*, and identified a potential ABRE in the sequence. In the transient expression analysis, it was observed that the CpAREB1 protein could activate the expression of ABRE-mini:GFP 4 times compared with control plasmid. This activation was dramatically decreased in the mABRE-mini:GFP reporter plasmids (Figure 5a,b). In addition, the protoplasts prepared from this protocol are also suitable for subcellular localization and protein–protein interaction assays, such as bimolecular fluorescent complimentary (BiFC), as described previously [55].

Taken together, these results demonstrate that we successfully established a useful transient expression platform based on an efficient protoplast preparation protocol in *C. pumila*.

### 2.3. A Universal Protoplast Preparation Protocol for Plant Biology Research

The wide application of scRNA-seq in plant biology is limited mainly due to the existence of rigid plant cell walls, which creates a challenge in dissociating the plant cells. In the light of the successful development of an efficient protoplast isolation protocol in *C. pumila*, we next asked if this method is applicable to multiple organs from diverse species. To this end, we conducted comparative studies of species representing key phylogenetic lineages in angiosperms. Following this protocol, abundant protoplasts were efficiently released from floral organs of *Aristolochia fimbriata* (tepals), *Aquilegia ecalcarata* (petal lobes), *Physalis floridana* (petals) and *Petrocosmea sinensis* (petal lobes), young fruits of *Glycine max* and *Lycopersicum esculentum*, tuberous roots of *Rehmannia glutinosa*, and gynophores of *Arachis hypogaea*. In some species, such as *Aristolochia*, it takes only three florets to generate an excessive number of protoplasts. In addition, cell integrity analysis using trypan blue staining shows that, similar to the situation in *C. pumila*, more than 95% of the protoplasts are living cells with an intact cell membrane system (Figure 6, for a higher resolution figure, please refer to Appendix A). Therefore, the protocol described here may break the limitation of broad usage of scRNA-seq in a non-model system and have potential to address the key issues in plant developmental and evolutionary-developmental (evo-devo) biology.

## 3. Materials and Methods

### 3.1. Plant Materials and Growth Conditions

*Chirita pumila* seeds were surface sterilized in 70% ethanol for 1 min followed by 2.5% sodium hypochlorite treatment for 3 min and rinsed five times in sterile water. The seeds were then germinated on 1/2 Murashige and Skoog (MS) basal medium supplied with 1% sucrose, 0.4% gelzan, and 0.02 mg/L α-Naphthalene Acetic Acid (NAA). The 4-week-old seedlings were transplanted on 1/2 MS medium in a glass tissue culture bottle at 26 °C under 10 h light/14 h dark conditions and light intensity of ~100 μmoL/m^2^/s in the growth chamber for 4–6 weeks [55].

*Aristolochia fimbriata*, *Aquilegia ecalcarata*, *Physalis floridana*, *Petrocosmea sinensis*, *Glycine max*, *Lycopersicum esculentum*, *Rehmannia glutinosa*, and *Arachis hypogaea* were grown under standard conditions in the glasshouse of the State Key Laboratory of Systematic and Evolutionary Botany, Institute of Botany, Chinese Academy of Sciences.

### 3.2. Plasmid Construction

pHBT-sGFP(S65T)-NOS plasmid [19] was used for the transformation efficiency evaluation and further downstream transient expression analysis. For constructing the plasmids for transient gene expression assay, the −46 bp CaMV 35S minimal promoter was synthesized and inserted into the pHBT-sGFP (S65T)-NOS plasmid by replacing the full length 35S promoter to generate the p35Smini:GFP control plasmid. The regulatory sequence (~3400 bp upstream the ATG) for the *CpTOE3* gene was isolated by TAIL-PCR. The ABRE on the *CpTOE3* gene were annotated using PlantCARE platform [56]. The core ABRE and the corresponding mutated version were repeated 4 times, synthesized, and inserted upstream of the CaMV35S minimal promoter of p35S mini:GFP plasmid to construct the reporter plasmids. For the construction of the regulatory effector plasmid and the transfection control plasmid, full-length AREB1 and β-glucuronidase (GUS) were isolated from *C. pumila* leaf cDNA and the pCAMBIA1301 vector, respectively, and inserted downstream of the CaMV 35S promoter in the pHBT-sGFP (S65T)-NOS plasmid by replacing the GFP. All primers used in the plasmid construction process are listed in Appendix A.

### 3.3. Protoplast Isolation and Transformation

A detailed protocol on how to prepare reagents and undertake protoplast isolation and transient expression are provided in the Appendix A.

### 3.4. Microscopy and Bioimage

The plant samples of each species were collected and recorded photographically with a Nikon D610 camera with a 105 mm prime lens. For protoplast observations, 10 μL cells were prepared on a slide and observed with a Leica DM68 microscope.

Confocal microscopy was performed on a Zeiss LSM-510 laser scanning microscope. GFP and FDA were excited at 488 nm wavelengths.

### 3.5. RNA Extraction and Expression Analysis

The protoplasts subjected to RNA extraction were collected by centrifuging at 250 rcf (g) for 3 min. Total RNA of protoplasts and unprocessed leaves were extracted using SV Total RNA Isolation System (Promega, Madison, WI, USA) and DNase I was added to digest the genomic DNA following the manufacturer’s instructions. Next, 1 μg of total RNA was reverse transcribed into cDNA using a RevertAid First Strand cDNA Synthesis Kit (Thermo, Waltham, MA, USA) according to the manufacturer’s instructions.

For real-time qPCR, ~120–200 nt gene-specific probes were designed and verified by standard PCR and sequencing. The efficiency of the primers (95–105%) was determined by creating a standard curve. The SYBR Premix Ex Taq (TaKaRa, Kyoto, Japan) was used to perform real-time qPCR with ROX as a reference dye on a StepOne Plus Real-Time PCR System (Life Technology, Carlsbad, CA, USA). The Ct value of each gene was determined by normalizing the fluorescence threshold. The relative expression level of the target gene was determined using the ratio = 2^−ΔΔCt^ method, and GUS was used as an internal control for protoplasts resulting from transient expression analysis and *CpACTIN* was used as a reference gene for expression analysis [55,57]. All primers used for gene expression analysis are listed in Appendix A.

### 3.6. Phylogenetic Analysis

The sequences were aligned using Clustal X software [58] and adjusted manually with the software Geneious version 7.1.4 [59]. Phylogenetic trees were constructed based on the Neighbor-Jointing (NJ) method [60] with MEGA 5.0 software [61]. Bootstrap values were estimated (with 1000 replications) based on Kimura’s 2-parameter model [62]. The accession numbers of the genes used in the phylogenetic analysis are listed in Appendix A.

### 3.7. Quantification and Statistical Analysis

All data are presented as means ± SD specified along with sample sizes (n) in the methods and in figure legends. Comparisons between groups for the analysis of qRT-PCR were performed with Microsoft Excel Student’s *t*-test, and significance levels are marked as: * *p* < 0.05, ** *p* < 0.01, *** *p* < 0.001, n.s., non-significant, *p* > 0.05.

## Figures and Tables

**Figure 1 ijms-23-03419-f001:**
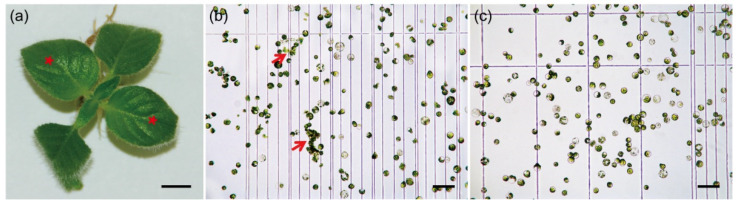
Protoplast isolation from *C. pumila* leaf mesophyll. (**a**) Young leaves (red stars) that were experiencing ongoing active growth were used for protoplasts isolation. (**b**) Protoplasts isolated using enzymatic buffers containing pectinase. The red arrows indicate incomplete digested tissues. (**c**) Protoplast isolated with pretreatment procedure and enzymatic buffers containing pectinase. Scale bars, (**a**) 0.5 cm; (**b**) and (**c**) 100 μm.

**Figure 2 ijms-23-03419-f002:**
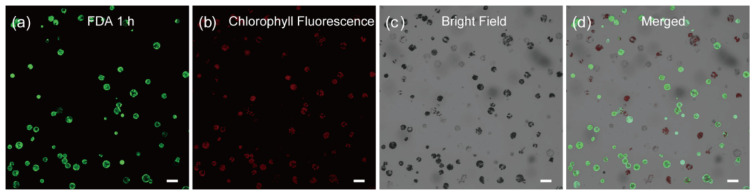
Viability assay of *C. pumila* mesophyll protoplasts after 1 h FDA staining. Protoplasts stained with FDA were imaged under (**a**) GFP, (**b**) chlorophyll fluorescence, (**c**) bright field, and (**d**) merged channel, respectively. The active cells are reflected by the green fluorescence signals. Scale bars, 50 μm.

**Figure 3 ijms-23-03419-f003:**
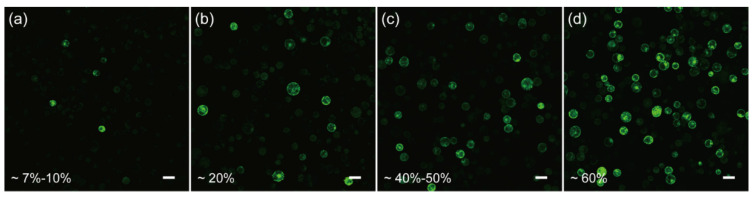
Optimization of the transient transformation system of *C. pumila* mesophyll protoplasts. (**a**) Under conventional PEG4000-mediated transformation, the efficiency was estimated around 7–10%. (**b**) PEG4000-mediated heat shock transformation improved the efficiency to 20%. (**c**) The transformation efficiency was improved to 40–50% when incubated in W5 solutions. (**d**) On the basis of (**c**), the transformation efficiency was further enhanced to 60% with 0.2 M CaCl_2_ in the PEG solution. Scale bars, 50 μm.

**Figure 4 ijms-23-03419-f004:**
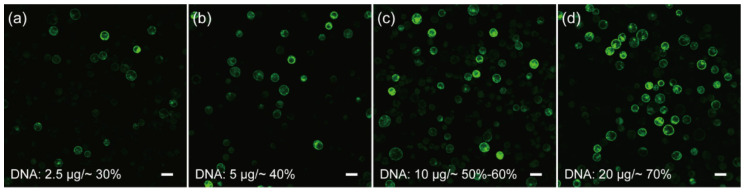
Influences of plasmid amounts on the transformation rates of *C. pumila* mesophyll protoplasts. Protoplasts transformed with (**a**) 2.5 μg results in an efficiency of ~30%, (**b**) 5 μg, (**c**) 10 μg, and (**d**) 20 μg plasmids with corresponding transformation efficiencies of ~40%, ~60%, and ~70%, respectively. Scale bars, 50 μm.

**Figure 5 ijms-23-03419-f005:**
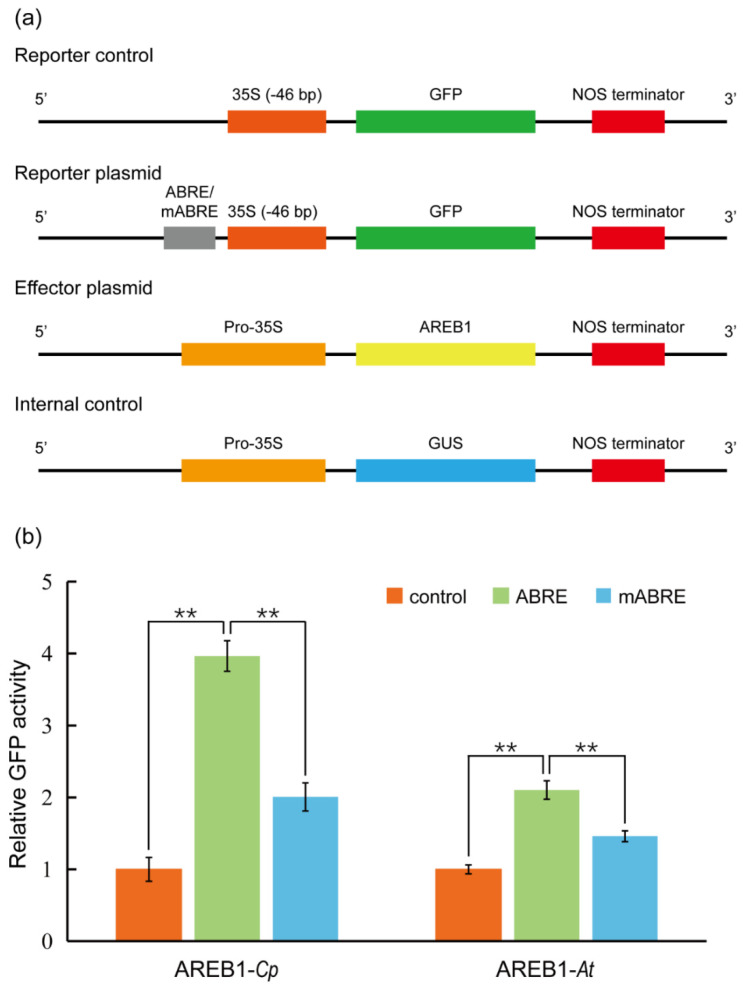
Transient gene expression analysis using *C. pumila* mesophyll protoplasts. (**a**) Schematic diagrams of plasmids used in the transient transactivation assays. The GFP driven by CaMV35S minimal promoter 35S (−46 bp) was used as a control. Four tandem repeats of ABRE or mutated ABRE(mABRE) fused with CaMV35S minimal promoter were used as reporter plasmids. AREB1 and GUS driven by CaMV35S promoter were used as effector plasmid and internal control, respectively. (**b**) AREB1 could activate the expression of the GFP reporter and the activation effect was significantly abolished when the core ABRE was mutated. Similar results were observed from the assays in *Arabidopsis* leaf protoplasts. The relative expression level of GFP was normalized to the internal control of GUS expression. The values shown are the average of three biological replicates. The statistical significance test was performed with Student’s *t* test (**, *p* value < 0.01).

**Figure 6 ijms-23-03419-f006:**
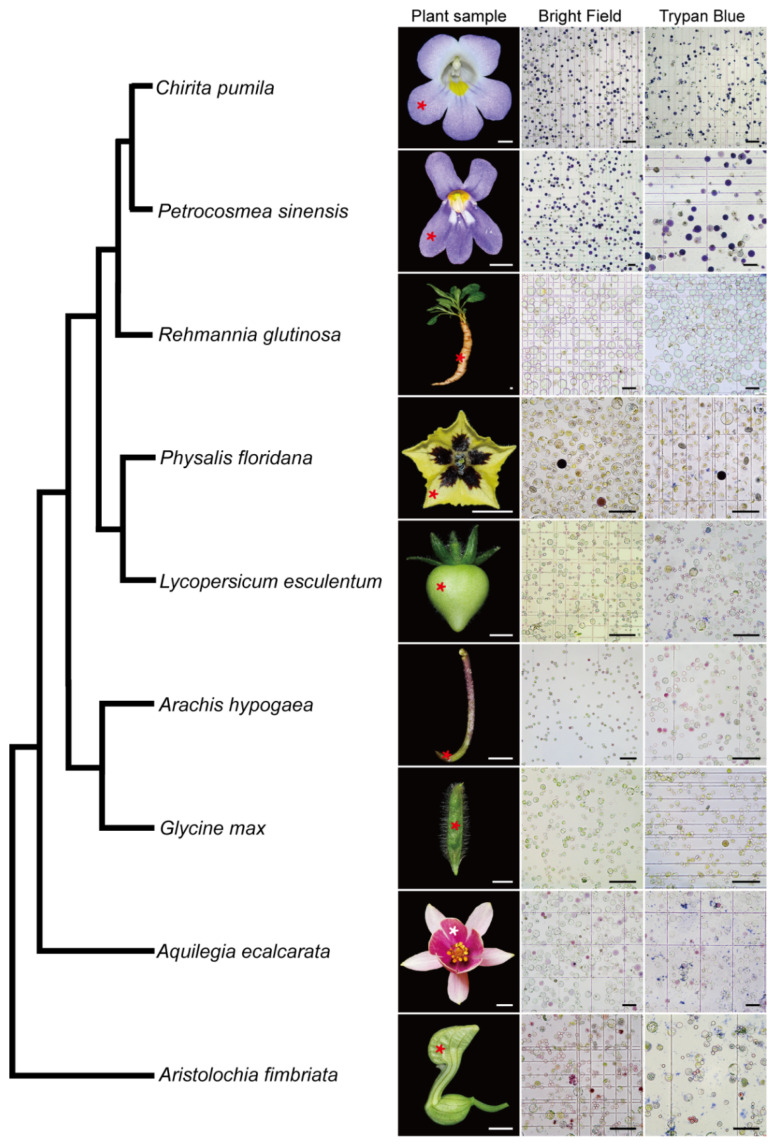
Validation of the protoplast isolation protocol from representatives of the angiosperm. Please note that some of cells are in dark purple or magenta color due to anthocyanin biosynthesis in the samples. The phylogenetic relationship of the representative species was based on Angiosperm Phylogeny Group (APG) IV. Asterisks in the plant sample row indicate the tissues used for the protoplast extraction. Scale bars, plant samples, 0.5 cm; protoplast images, 100 μm.

**Table 1 ijms-23-03419-t001:** Effects of different enzyme combinations on protoplast yield from *C. pumila* leaves.

Cellulase (%)	Macerozyme (%)		Pectinase (%)	
		0.2	0.5	1
0.5	0.2	0.5 ± 0.06	2.9 ± 0.23	3.0 ± 0.28
	0.5	1.6 ± 0.12	5.3 ± 0.33	4.5 ± 0.18
	1	3.0 ± 0.21	4.9 ± 0.35	3.4 ± 0.17
**1**	0.2	0.8 ± 0.11	5.3 ± 0.43	5.4 ± 0.37
	**0.5**	2.0 ± 0.17	6.8 ± 0.45	4.7 ± 0.21
	1	3.5 ± 0.37	6.2 ± 0.37	3.5 ± 0.13

The optimal enzyme combinations are highlighted by bold letters. The numbers indicate the average protoplast yield (×10^5^ cell/gFW) from three replicates with the standard error shown in the table.

## Data Availability

This study did not generate any unique datasets or code.

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
