# Peer review of "An Efficient and Universal Protoplast Isolation Protocol Suitable for Transient Gene Expression Analysis and Single-Cell RNA Sequencing"

_ijms, 2022, doi:10.3390/ijms23073419_

Round 1
Reviewer 1 Report
herein authors describe a procedure for cell wall removal and protoplast isolation in specimen of the genus Chirita pumila.
Experimental section is confusing:
-description of RNA extraction from protoplast is described first than protoplast preparation;
-amplicon lenght for qPCR need to be defined;
-list of reagents and experimental procdures need to be better integrated.
Additionally, the authors describe such protocol as universal, but the authors defined it in Lamiacee; thus, other plants need to be tested for a "universal" procedure.
Author Response
- description of RNA extraction from protoplast is described first than protoplast preparation;
Response: Thank you for pointing this out. We have adjusted the orders of RNA extraction and protoplast preparation, see Page 9, Line 242-245 and Line 252-256.
- -amplicon lenght for qPCR need to be defined;
Response: Thanks for your suggestion. We have defined the length of qPCR probes in the “Materials and Methods” section (Page 9, Line 259).
- -list of reagents and experimental procdures need to be better integrated.
Response: Thank you for this good suggestion. We have moved the experimental procedures into the supplemental file and reorganized the list of reagents and experimental procedures in the revised manuscript (Please see the Supplemental Method for detail).
- Additionally, the authors describe such protocol as universal, but the authors defined it in Lamiacee; thus, other plants need to be tested for a "universal" procedure.
Response: Good point! Actually, we tested this protocol in different samples from diverse plant species representing the key branches in angiosperms, please see the “Results and Discussion section 2.3 and the associated figure 6” (Page 7, Line 191-207).
Reviewer 2 Report
Dear Authors,
I have a great honor to review manuscript entitled:” An efficient and universal protoplast isolation protocol suitable for transient gene expression analysis and single-cell RNA sequencing” which is considered for publication in IJMS. The article is good written and present interesting methodological approach for protoplast isolation. Manuscript need improvements before acceptance whhich I present below in form of list:
Article has not formulated precisely aim of the study and any hypothesis. A Introduction must ended with this elements
Figure 2 -Figure 4. I strongly recommend to change photos to photos from higher magnification. Now the magnification is to low to clearly observed protoplast itself.
On Figure 6 photos frm Bright field and Tryptan blue must be separated from phylogenetic tree. Because, now the photos itself is so small than any differences could not be spotted between different plants.
The point section (Materials&Methods) 3.5 and whole 3.6 should be moved to the supplementary data. Now the way ow presentation is suit well into methods description in IJMS standards. However this information is crucial for procedure it will be ideal for supplementary data.
Moreover the article need point by point characterized methodological approach for statistical analyses of results. Currently manuscript (Materials&Methods section) did not have clearly presented statistical analyses description.
Sincerely,
Author Response
1. Article has not formulated precisely aim of the study and any hypothesis. A Introduction must ended with this elements
Response: Thank you for this excellent suggestion, we have revised the MS by adding a short and precise aim of this study in the introduction section (Please see Page 2, line 52-60).
2. Figure 2 -Figure 4. I strongly recommend to change photos to photos from higher magnification. Now the magnification is to low to clearly observed protoplast itself.
Response: Thank you for this your suggestion. Following your suggestion, we have repeated the protoplast preparation process and took imagers with higher resolutions and substitute them in figure 1.
3. On Figure 6 photos frm Bright field and Tryptan blue must be separated from phylogenetic tree. Because, now the photos itself is so small than any differences could not be spotted between different plants.
Response: Thank you for this comment. In the work, we tried to universalize the protocol developed in C. pumila onto angiosperms by selecting representatives from the key branches. In the light of evolutionary relationships and make it evident for comparisons, we think it is better to keep the phylogeny and protoplast together as a whole. The main information conveyed by this figure is that the protocol developed in C. pumila is universal and when applied in other plant species, high-quality protoplast will be effectively released similarly. We therefore prefer to keep the main figure as it is. However, in the revised MS, we incorporated all the images with high resolution in the Supplemental Figure 5 if readers want to check the protoplast in detail.
4. The point section (Materials&Methods) 3.5 and whole 3.6 should be moved to the supplementary data. Now the way ow presentation is suit well into methods description in IJMS standards. However this information is crucial for procedure it will be ideal for supplementary data.
Response: Indeed, this is a very good suggestion. We have moved the whole detailed step-by-step methods into the supplemental files as “Supplemental Method” in the revised manuscript.
5. Moreover the article need point by point characterized methodological approach for statistical analyses of results. Currently manuscript (Materials&Methods section) did not have clearly presented statistical analyses description.
Response: Thank you very much for raising this concern. We are sorry for not providing the details of the statistical analyses of results. In the revised MS, we have added a section “Quantification and statistical analysis” in the materials and method section.
Reviewer 3 Report
Potentially intersting protocol, but require significant impovemnet in the text and discussion.
Below are some points:
Line 63: what is highly active?
Line 64: we - must be We.
Here you mention sc-RNA-seq method, but in the case the key points is to keep gene expression and chromatin status in the same stage as in native cell. However, this task does not fit with heat-shock and whole procedure of pp isolation you used. You need to prove that your steps, buffers, pH, heat-shock did not alter epigenetic and gene expression in the cells.
Line 69: tissue innovations??
Lione 91: „significantly increased with this treatment“ – this is not visible from figure 1. Quantitative data like rhd123 signal or else are required. Moreover, so many cell look like damage in both cases,
Figure 1: please, provide legends, not descriptions of the results. Description of the results must be in results.
Line 97: „fluorescein diacetate (FDA) staining“ in not a viability and not activity. It can be only proteases activity.
Figure 2: please, increase quality of the images. Panel (a) – better named as FDA, not GFP. Moreover, FDA is not a viability.
Line 184: „cell viability analysis using trypan blue staining shows that, similar to the situation in C. pumila, the protoplast exhibits extreme high activities (more than 95%; Figure 6).“ – trypan blue is not a activity. Please, re-formulate.
Line 254: Calcium chloride itself does not precipitate.
Line 266: this point is not neccesary here: „Critical: MES is very prone to be contaminated by bacterial and mold, please conduct this step in a laminar hood. The MES solution is critical for a successful experiment.“.
Line 329: vacuum quality is a key, please, provide vacuum kinetics.
Line 337: „Fix the petri dish on a shaker at a speed of 40 rpm and 26 ˚C for 3 hrs.“ This step lead to protoplasts damage and serve as additional mechanical stress what in turn altered gene expression.
Despite W5 is a good for stabilization of the cell mebrane and protpplasts stability, this buffer contain huge amount of chloride and sodium what have significant effect on gene expression. More natural medium are require for sc-RNA-seq investigation. Please, discuss this point.
Line 389: „2 μL1 mg/ml fluorescein diacetate (FDA)“? Please, convert to µM and provide details. It is very clear that FDA here is not reflect cell viability. FDA can penetrate in the cell with membrane damage and haydrokesed to fluorescein and proteases. If you want to FDA, it is necessary to monitor fluorescein leakage from the cells during 1-2 hours ta least, of you rhodamine 123 (https://www.mdpi.com/2223-7747/10/2/375/htm).
Line 453: please, convert Lux to µmol.
Line 489: 600 lex??
„Cultivation“ of protoplasts in the W5 buffer for 24 h lead to signioficant chnages in natural gene expression and resulted cell can not be consuidered as representative from original tissue. You can not extrapolate data to original cell in planta.
In planta gene expression regulated by hormonal signaling, carbohydrates, ions.
In your case cells are 24 hours or more in the buffers without hormones, ions, carbohydrates. How can you linked these cells with cells in planta?
Author Response
1. Line 63: what is highly active?
Response: We are sorry for the confusing information. We used “highly active” to express the protoplast are active with strong competent capacity. In the revised MS, we have changed “highly active and transformable” into “competent for transformation” (Page 2, Line 63-64).
2. Line 64: we - must be We.
Response: Thank you! We have corrected this typo (Page 2, Line 65).
3. Here you mention sc-RNA-seq method, but in the case the key points is to keep gene expression and chromatin status in the same stage as in native cell. However, this task does not fit with heat-shock and whole procedure of pp isolation you used. You need to prove that your steps, buffers, pH, heat-shock did not alter epigenetic and gene expression in the cells.
Response: Thank you for this good suggestion. It is very true that buffer treatment in digestion process has the potential to incur gene expressional changes, especially the stress-responsive genes. For a success sc-RNA-seq experiment, a sister RNA-seq analysis, comparing the isolated protoplast and unprocessed tissues is essential to identify the genes induced by protoplasting process. These genes have to be filtered out and is vital to mitigate the impact of global transcriptomic changes on the scRNA-seq data (see Zhang et al., 2021 Dev. Cell, 56: 1056-1074 as an example). In the revised manuscript, we conducted additional q-PCR analysis based on the RNAs extracted from the integral leaves and protoplast prepared using this protocol. We show that the expression of key components involving in the histone modifications (HDA6, HDA19, IBM1, SWN and CLF) was not significantly changed upon digestion, suggesting the general epigenome is not altered in the digested cells compared with the whole leave. In the revised MS, we have included these data and made a short discussion (Page 3, Line 98-110 and Supplemental Figure 3). Regarding to the reviewer’s concern about the heat-shock, we don’t think it is a major factor in the protoplasting process, as it is a primary determinant in the protoplast transformation process.
4. Line 69: tissue innovations??
Response: We are sorry for the confusing information. We used “tissue innovations” to express the tissues with novel function or morphology in evolution. In the revised manuscript, we have rephrased this sentence as “…underlying the morphological novelties” (Page 2, Line 70).
5. Lione 91: „significantly increased with this treatment“ – this is not visible from figure
6. Quantitative data like rhd123 signal or else are required. Moreover, so many cell look like damage in both cases,
Response: Thank you for this suggestion. In the revised manuscript, we have repeated the digestion process and evaluate the quality and quantity of the protoplast isolated with and without pretreatment procedure. The data show that the number of high quality cells increased from ~78% to ~93% along with the pretreatment. In the revised MS, we have incorporated these new data to support our conclusion (Page 2, Line 92-93). Also, we have taken pictures with higher resolution and magnification to support our conclusion (Page 3, Figure1 and Supplemental Figure 1).
6. Figure 1: please, provide legends, not descriptions of the results. Description of the results must be in results.
Response: Thank you for this good suggestion. We have rephrased the figure legend thoroughly to make it more accurate instead repeating the results.
7. Line 97: „fluorescein diacetate (FDA) staining“ in not a viability and not activity. It can be only proteases activity.
Response: We are sorry for not mentioning FDA staining in detail. Fluorescein diacetate (FDA), freely enters living cells where it may be rapidly hydrolyzed by enzymes to give fluorescein (Pritchard, 1985, Plant, Cell and Enviro., 8: 727-730). In cell biology studies, FDA-staining, has been adopted as a conventional way to detect cell viability (Please see mammalian cell studies in Wilkesmann et al., 2020, Methods & Proto., 3:30 and Klak et al., 2021, Micromachines, 12: 304; Plant cell studies in Lei et al., 2015, MethodX, 2:24-32 and Impe et al., 2020, Front Plant Sci., 10: 1588). In the revised manuscript, we have rephrased this sentence as “…more than 95% of the cells are viable with an integral cell membrane system, as illustrated by strong fluorescent signals.” (Page 3, Line 115-116).
8. Figure 2: please, increase quality of the images. Panel (a) – better named as FDA, not GFP. Moreover, FDA is not a viability.
Response: We are sorry for not providing the high-quality images in the previous MS. In the revised MS, we have repeated the FDA staining with different concentrations and substitute the figure 2 with better images.
9. Line 184: „cell viability analysis using trypan blue staining shows that, similar to the situation in C. pumila, the protoplast exhibits extreme high activities (more than 95%; Figure 6).“ – trypan blue is not a activity. Please, re-formulate.
Response: Thank you for pointing this out. In the revised MS, we have rephrased this sentence as “…cell integrity analysis using trypan blue staining shows that, similar to the situation in C. pumila, more than 95% of the protoplast are living cells with an intact cell membrane system,…” (Page 7, Line 202-205).
10. Line 254: Calcium chloride itself does not precipitate.
Response: Thank you for pointing this out. In the revised MS, we have rephrased this note as “Calcium chloride should be sterilized by filtering.” (See Supplemental Method).
11. Line 266: this point is not neccesary here: „Critical: MES is very prone to be contaminated by bacterial and mold, please conduct this step in a laminar hood. The MES solution is critical for a successful experiment.“.
Response: Thank you for this suggestion. In the revised MS, we have removed this notion.
12. Line 329: vacuum quality is a key, please, provide vacuum kinetics.
Response: Thank you for this suggestion. In the revised MS, we have add a notion for the vacuum kinetics (Please See Supplemental Method).
13. Line 337: „Fix the petri dish on a shaker at a speed of 40 rpm and 26 ˚C for 3 hrs.“ This step lead to protoplasts damage and serve as additional mechanical stress what in turn altered gene expression.
Response: Thank you for raising this concern. Actually,40rpm is a very low speed and is good for an efficient hydrolytic reaction. According to the experimental data we obtained, we didn’t observe significant cell damage at this speed for protoplasting. Additionally, for a sc-RNA seq experiment, a control experiment to filter the protoplasting-responsive genes is necessary. For more detail, please see our response to point 3.
14. Despite W5 is a good for stabilization of the cell mebrane and protpplasts stability, this buffer contain huge amount of chloride and sodium what have significant effect on gene expression. More natural medium are require for sc-RNA-seq investigation. Please, discuss this point.
Response: Thank you for raising this concern. We fully agree with the reviewer’s opinion that chloride and sodium buffer could potentially stimulate global expressional change upon long-time treatment. However, these differentially expressed genes resulted from the digestion process will all be filtered by the control experiment in the sc-RNA seq. In the revised MS, we have added a short discussion about this issue (Page 3, Line 98-104).
15. Line 389: „2 μL1 mg/ml fluorescein diacetate (FDA)“? Please, convert to µM and provide details. It is very clear that FDA here is not reflect cell viability. FDA can penetrate in the cell with membrane damage and haydrokesed to fluorescein and proteases. If you want to FDA, it is necessary to monitor fluorescein leakage from the cells during 1-2 hours ta least, of you rhodamine 123 (https://www.mdpi.com/2223-7747/10/2/375/htm).
Response: Thank you for raising this concern. FDA concentrations are usually use as unit of mg/mL, and a final concentration of 0.1mg/mL is the optimal working concentration. In order to dynamically quantify the cell membrane integrity and hence the potential of signal leakage of FDA, we monitored the fluorescence signal from the protoplast stained with a gradient of 0.2mg/mL, 0.5 mg/mL, 0.8 mg/mL, 1 mg/mL and 2 mg/mL FDA within 10 mins. We found a consistent fluorescence signal appearance under different FDA concentration, suggesting FDA concentration doesn’t make a big difference in detecting the cell membrane integrity. In the revised manuscript, we have incorporated these new data and made a short discussion (See Page 3, Line 114-118 and Supplement Figure 4).
16. Line 453: please, convert Lux to µmol.
Response: Thank you for this suggestion. We have converted 600 Lux to ~ 11 µmol/m2/s in the revised MS (See Supplemental Method).
17. Line 489: 600 lex??
Response: Thank you. We have converted 600 Lux to ~ 11 µmol/m2/s in the revised MS, also we have systematically checked the MS to avoid any incongruence about the light intensity unit (See Supplemental Method).
18. „Cultivation“ of protoplasts in the W5 buffer for 24 h lead to signioficant chnages in natural gene expression and resulted cell can not be consuidered as representative from original tissue. You can not extrapolate data to original cell in planta.
Response: Thank you for raising this concern. We cultivate the cells in the W5 for 24 hours for the transient expression analysis instead of sc-RNA seq analysis, these experiments were conducted to verify the applicable of the protoplast for gene expression analysis, such as sub-cellular localization, protein-protein interaction, protein-promoter interaction, ect. For preparing the protoplast for sc-RNA seq, the cells after two-round digestion are immediately ready for library preparation. For your concern about extrapolate data to the original cell in planta, there is conventional pipeline on how to map the cells back to the tissue/organ in the data analysis process of sc-RNA seq (See discussions in Birnbaum, 2018, Annu. Rev. Genet., 52, 203–221; Jha et al., 2021, eLife, 10, e66877).
19. In planta gene expression regulated by hormonal signaling, carbohydrates, ions. In your case cells are 24 hours or more in the buffers without hormones, ions, carbohydrates. How can you linked these cells with cells in planta?
Response: Thank you for raising this concern. For details, please see our response to point 18.
Round 2
Reviewer 1 Report
The authors included elements that positively affect the quality of manuscript
Author Response
The authors included elements that positively affect the quality of manuscript
Response: Thanks so much for your suggestions to improve the quality of our manuscript.
Reviewer 2 Report
Dear Authors,
All my suggestions was adressed and corrcted I recomend publication.
Sincerely,
Author Response
Dear Authors,
All my suggestions was adressed and corrcted I recomend publication.
Sincerely,
Response: Thanks so much for your suggestions
Reviewer 3 Report
Plesae, see also attached file.

Author Response
The paper looks more sound, but still some points are missing.
- Lines 92-93: FDA can enter cell with partially membrane integrity and to death cells as well. Among enzymes what can hydrolized FDA are also proteases, lipases, DNAse etc. (DOI: 10.1016/j.soilbio.2005.06.020; https://pubmed.ncbi.nlm.nih.gov/11438191). The most suitable way of FDA usage is to study fluorescein retention time in kinetics (after 2-3 hours after loading, for exmaple). This allow you to really detect viable cells. You can find some details here: https://doi.org/10.3390/plants10020375.
Response: Thank you for the good suggestion. Following your suggestion, we have repeated the protoplast activity check using FDA and Rhodamine 123 stanning in time-laps kinetics. see Page 4, Line 137 and supplemental figure 4 and 5.
- From this point of view, Rhodamine 123 is much reliable dye. Figures 2, S4: it will be nice to increase brightness in the chlorophyll panel.
Response: Thank you for this comment. We have conducted additional experiment using Rhodamine 123 as an indicator for cell viability and included the new results in the Supplemental Figure 5 and text, Page 4, Line 137.
- Moreover, it looks like you did not estimate cell viability correctly. In figures 2 and S4, I label cell that is sure, not viable: protoplasts are round with clear visible cytoplasm. But these cells are not round and look like semi-death. Maybe they have some proteases that can give you green fluorescence. You need to evaluate your images very carefully and re-calculate, or use another dyes.
Response: Thank you for raising this concern. We have repeated the FDA staining following your suggestion with time-lapse images. We found that after 3h staining, the cell viability rate is 86.89%, and this number is further evidenced by the Rhodamine 123 staining. We have incorporated these updated results in the revised manuscript. Please see Page3, Line 119.
4.Line 98-110: these results is very inetersting, because does not fit with previously published papers in which wounding induced significant epigenetic modification even after 1 hour: :https://academic.oup.com/plcell/article/12/5/707/6008817?login=true
https://www.nature.com/articles/s42003-019-0646-5
In addition, protoplasts isolation significantly chnage chromatin chromatin topology, nuclei flatness, chormocenter distribution, nucleoli size, what, in turn have a significant effects on epigenetic status. This issue need to be discussed.
Response: Thank you for the good suggestion. We have added a short discussion about the expression analysis of executive factors involved in the histone modification machinery. Please see Page3, Line112-115.
5.Line 115: „integral cell membrane system“ – but FDA fluoresence itself nothing to do with membrane intergrity. Membrane intergrity is reflected by FDA leakage kinetics.
Response: Thank you for pointing this out. We have revised this statement based on our new data. Please see Page3, Line 118-123.
6.Line 222: please, add light intensity.
Response: Thank you for the suggestion, we have added the details about the light intensity.
7.Line 223: it is very interesting, but did not discuss in the paper that Chirita pumila have a completely different nutrient requirements as other species. In the soil (glasshouse/greenhouse) NPK ratio is 5:1:3, while for Chirita pumila you used 60:1.25:40. In addition, the major anion in your case is Cl what, in turn, prevent phosphate uptake. This made ratio even worse. Please, explain this fact in few sentences in discussion.
Response: Thank you for raising this concern. For the growth of Chirita pumila seedlings, we use 1/2MS medium supplied with 1% sucrose, which is a standard growth medium for an array of plants.
8.Figure 6: please, check the scale bar and images carefully. It looks like in many cases, image were confused between trypan blue and dic image. Please, add the correct scale bar.
Response: We are sorry for not checking the scale bar carefully. In the revised MS, we have corrected the scale bars in Figure 6 and other figures as well.
9.Line 285: please, clarify what do you mean by quality: freshly isolated or after some time? The method you have used? Do you mean protease/esterase activities or membrane integrity?
Response: Thank you for raising these questions. We have added an explanation on how to evaluated the protoplast quality in the figure legend of Figure S 1.
Round 3
Reviewer 3 Report
Thank you for the corrections.
It will be also important to compare chromatin topology in original explants and in the protoplasts.
https://www.biorxiv.org/content/10.1101/2021.11.26.470128v1.abstract
Line 223: the question is not about the name of the medium, but about composition. How nutrional stress you apply to Chirita pumila affected gene expression, cell wall thickness, possible ploidy level etc.
The other species have no nutrional stress.
Supplemental figure:
Line 44: You can use DMSO which in concentration 1 ml/l does not have effect on membrane stability. Reduction in the number of FDA positive cell maybe related with low protoplasts quality, not with acetone.
Line 57-59: it seems that authors confuse rhodamine 123 (rhd123) meaning. Rhd123 mean membrane potential and authors clearly show that protoplasts gradually recover activity from 0.5 to 12 h and positive rhd123 mean high cell metabolic activity.
Please, have a look: https://www.mdpi.com/2223-7747/10/2/375/htm
Supll. Fig 6: plesae, make corrections.
Suppl fig 7: first row seems to be incorrect: all images like trypan blue.
Other rows: it look like cell are very heterogenous with very different cell types, ploidy and seems to be gene expression profile what made results of gene expression very questionable. There are several tricks what allow to isolate only one cell type and rich homogenous populations.
Supplementary M&M:
Line 26: I am not sure that you can not autoclave CaCl2.
Line 58: Which enzyme activation?? 55°C has been used to deactivate proteases. Not for enzyme activation.
Line 59: please, remove one point.
Please, describe this step more precisely: either you adjust volume to 100 ml, or you add 1 ml H2O. Plesae, consider that enzyme also have some volume. If you mention like currently, the final volume will be more as 100 ml.
Line 70: it seems to be you use in this buffer only Cl as anion. Why? It is quite toxic fort he cell, it will be nice to use other anoin to make this solution more balanced. See link below.
Line 166-169: how did you dissolve FDA and rhd123? Acetone is no a suitable decision for several reasons, the best one is DMSO, please, consider that DMSO as the concentration 1ml/l does not have effect on cells.
Round 4
Reviewer 3 Report
Thank you for your response.
Please, consider that gene expression mean finally protein function, not RNA level. You made phospahet starvation for a long time (as one of example), so, it may mean that phosporylation as main regulator of protein activity can significantly changes, but mRNA level keep the same level.
It is definitely a very good idea to perform chromatin topology in the future.
Medium composition:
Nutrion is a key factor in plant developmnet, Do you claim that C. pumila require N:P:K ratio 60:1.25:40? What mean high relative phosphate deficeincy, Cu-deficeoncy and dso on.
This is normaly consider as very high nutrional stress.
Cell heterogenity: the main goal oft he protopalsts isolation is to get cell-type specific gene expression profile. Gene e0xpression, indeed, were very different in vasculature, mesophyll cell, epidermis (in the case of the leaf). Combination of the all cell types for RNA isolation have much less sense as single cell type.
For the W5:
If even some people use toxic chloride as major anion, avoiding phosphate, sulfur and other, does not mean that this is good idea. In W5 occurred quite rapid cell degradation, induction of lytic vacuole, down regulation gene expression (at the protein functional level).
Please, have a look: https://www.mdpi.com/2223-7747/10/2/375/htm
Plesae, correct typos (doble point at line 59) etc.
Author Response
Thank you for your response.
- Please, consider that gene expression means finally protein function, not RNA level. You made phosphate starvation for a long time (as one of example), so, it may mean that phosphorylation as main regulator of protein activity can significantly changes, but mRNA level keeps the same level.
Response: Thank you for the good suggestion. Yes, the reviewer is correct, the RNA level doesn’t equal to protein function, especially when the tissues are stressed by abiotic factors. In the revised MS, we have paid special attention when interpretation of our data.
- It is definitely a very good idea to perform chromatin topology in the future.
Response: Thank you for your agreement.
- Medium composition: Nutrition is a key factor in plant development, Do you claim that C. pumila require N:P:K ratio 60:1.25:40? What mean high relative phosphate deficiency, Cu-deficiency and so on. This is normally considered as very high nutritional stress.
Response: Thank you for raising this concern again. In the manuscript we didn’t mention the requirement for NPK ratio for the growth of C. pumila nor mentioned the details of NPK ratio in the material and method section. The original description for the growth condition of C. pumila is as follows “The seeds were then germinated on 1/2 Murashige and Skoog (MS) medium supplied with 1% sucrose, 0.4% agar and 0.02 mg/L α-Naphthalene Acetic Acid (NAA). The 4-week-old seedlings were transplanted on 1/2 MS medium in a glass tissue culture bottle at 26 ˚C under 10 hrs light/14 hrs dark conditions and light intensity of ~100 μmoL/m2/s in the growth chamber for 4-6 weeks.”
- Cell heterogeneity: the main goal of the protoplast isolation is to get cell-type specific gene expression profile. Gene expression, indeed, were very different in vasculature, mesophyll cell, epidermis (in the case of the leaf). Combination of all the cell types for RNA isolation have much less sense as single cell type.
Response: Thank you for raising this concern. The reviewer is correct that different cell type has distinct gene expression profiles. The differential gene expression in different cell types sets the foundation to group and map the cells back to the position or organization in the tissue in the scRNA-seq experiment and to generate the gene expression landscape. The aim of this study is to establish a general and optimized protoplast preparation protocol for multiple organs from distinct species. Therefore, cell heterogeneity in the protoplast suspension is a good indication of successful isolation of different cell types from the tissue of interest.
- For the W5: If even some people use toxic chloride as major anion, avoiding phosphate, sulfur and other, does not mean that this is good idea. In W5 occurred quite rapid cell degradation, induction of lytic vacuole, down regulation gene expression (at the protein functional level). Please, have a look: https://www.mdpi.com/2223-7747/10/2/375/htm
Response: Thank you for raising this concern. The W5 buffer is a widely used reagents in the plant cell biology studies and we checked the literatures, which shows a consistent composition with minor difference in the MES concentration (e.g. Yoo et al., Nat. Protoc. 2007, 2, 1565-1572; Mazarei et al., Biotechnol. J. 2008, 3, 354-359; Sultana et al., Plant Cell Rep. 2019, 38, 1329-1345; Ren et al., Front Plant Sci. 2021, 12, 626015). For a scRNA-seq experiment, the W5 buffer is normally used to wash the cell debris and the protoplast will be immediately resuspended with 0.6M mannitol. In the current study, we incubated the transformed protoplast in W5 buffer for ~20 hours and we didn’t find a significant changes in the gene down-regulation in the transient gene expression analysis nor cell degradation in the GFP signal detection experiment.
Please, correct typos (double point at line 59) etc.
Response: Thank you for the good suggestion. We have checked the MS thoroughly to avoid any typos.
Here, we want to express our gratitude to the Reviewer 3 again for the careful review of our manuscript, all the comments and suggestions raised by the Reviewer 3 are very valuable and constructive. The manuscript has experienced a significant improvement following the suggestions and comments made by Reviewer 3.